# The Role of Telemedicine in the Treatment of Cognitive and Psychological Disorders in Parkinson’s Disease: An Overview

**DOI:** 10.3390/brainsci13030499

**Published:** 2023-03-16

**Authors:** Desirèe Latella, Giuseppa Maresca, Caterina Formica, Chiara Sorbera, Amelia Bringandì, Giuseppe Di Lorenzo, Angelo Quartarone, Silvia Marino

**Affiliations:** IRCCS Centro Neurolesi “Bonino Pulejo”, 98149 Messina, Italy

**Keywords:** telemedicine, parkinson’s disease, cognitive decline, telehealth, psychological support

## Abstract

Background: This literature review evaluates the use and efficacy of telemedicine in cognitive and psychological treatment in Parkinson’s disease. Methods: Studies performed between 2016 and 2021 that fulfilled inclusion criteria were selected from PubMed, Scopus, Cochrane, and Web of Science databases. All articles were evaluated by title, abstract, and text. All studies that examined the cognitive and psychological/psychotherapy treatment of patients with Parkinson’s disease by telemedicine were included. Results: Telehealth improved cognitive status and emotional/behavioral disorders in this population, and had positive effects on the patients’ and caregivers’ quality of life. Conclusions: Our literature review supports the development and efficacy of cognitive and psychological treatment with telemedicine, but the methodology of the study must be reviewed considering its limitations so as to highlight the benefits and risks of treatment via telemedicine.

## 1. Introduction

Parkinson’s disease (PD) is the second-most common neurodegenerative disorder. The primary cause of the disease is dopamine deficiency in the basal ganglia that leads to progressive movement deterioration, such as bradykinesia, tremor, rigidity, and later postural instability. Of those aged over 55 years, 1% are PD-affected, and the average age of onset is about 60 years. The prevalence of PD increases with age in those over 60 years old [1]. There are many theories about the pathogenesis of the disease. There are mutations in several genes that cause autosomal dominant or recessive forms of PD [2]. Many studies considered oxidative stress, metabolic changes, telomere shortening, dysfunction of cellular proteolytic and mitochondrial systems, or cardiovascular factors. Studies that lead back to the theory of neuroinflammation as the cause of PD seem to be more plausible, but it remains unclear if prolonged inflammation is the cause or an effect of neurodegeneration [3]. Chronic neuroinflammation and microglial activation play a role in PD neurodegeneration [4]. This process leads to blood brain–barrier damage and non-protection of the nervous system, permitting the infiltration of peripheral immune system chemokines. The result is an activation of glial cells, T cells, and mast cells that increase neuroinflammation with consequent neuronal death. This event improves the neurotoxic molecules causing neurodegeneration. Mutual activation of inflammation in the brain and peripheral immune cells leads to the release of neurotoxic molecules and exacerbates neurodegeneration [3]. This disease can also be associated with a wide spectrum of debilitating nonmotor symptoms due to nigral degeneration (apathy, dysphoria, cognitive impairment) and extranigral degeneration (hyposmia, sleep disorders, autonomic dysfunction, cognitive deficits, psychosis, depression, anxiety, and apathy) [5]. More than 24% of patients with PD have one or more cognitive symptoms [5]. The most affected cognitive functions are attention, language, and speech abilities, visuospatial memory, and executive processes [6], worsening quality of life [7]. The cognitive impairment increases the risk of developing dementia. Neuropsychiatric symptoms, such as depression (35%) and anxiety (31%) disorders, are observed in more than 60% of PD patients and are recognized as important predictors of patient disability [8], quality of life, and mortality. These symptoms may precede the onset of motor symptoms, even by years, and are independent of therapy, whereas others, such as psychosis and impulse control disorders (ICDs), are correlated with dopaminergic therapy [9]. The gold standard treatment is levodopa, which continues to be the most effective treatment for PD. Moreover, the use of levodopa allows an improvement in motor control, but can highlight neuropsychiatric disorders. The effectiveness of pharmacological interventions in treating PD neuropsychiatric symptoms is contrasting, as they cause unfavorable side effects, including exacerbation of motor problems. Consequently, psychological interventions provide a valid alternative treatment [10]. Early recognition and monitoring of cognitive and behavioral difficulties in PD are important to promote patient management and adequate rehabilitative strategies [11]. Concerning cognitive rehabilitation, task execution can be carried out in different modalities. In particular, it can be performed with pen-and-pencil exercises or innovative tools based on robotics and virtual reality. Computer-assisted cognitive rehabilitation improved specific cognitive domains such as attention, orientation, and visuospatial abilities [12]. Physical exercise was originally proposed as a PD treatment many years ago [13] and motor rehabilitation therapies were considered an adjuvant to pharmacological treatment to maximize functional abilities and improve quality of life. A recent meta-analysis of physiotherapy interventions showed significant and clinically important benefits for walking speed and balance in PD patients [14]. Fisher et al. documented neuroplasticity of dopaminergic signaling in D2 receptors in four patients with early-stage PD following a treadmill exercise [15]. Moderate-PD patients after balance training showed gray matter changes on voxel-based morphometry and this correlated with improvements in motor skills [16]. These studies demonstrated motor rehabilitation-induced brain connectivity changes and increased synaptic strength with a consequent improvement in quality of life. To be effective, cognitive–behavioral rehabilitation and psychotherapeutic treatment should be continuous over time, even if the SARS-CoV-2 pandemic reduced access to care facilities. This event forced people to socially distance and quarantine abruptly, interrupting access to routine medical care [16,17]. Patients with neurological diseases are frail due to advanced age, comorbidities, or immunosuppression due to treatments. Telehealth care is a successful method that uses information and communication technology (ICT), in which the health professional and the patient interact remotely [18]. In addition, telemedicine supports the patient in their rehabilitation process, allowing for improvements in the outcome and reducing national health-care system costs. The application areas of telemedicine are various and can include telecardiology, teledermatology, telestroke, and telerehabilitation [19]. Consequently, telemedicine became an integral part of nonemergency health care [17,18]. Studies have demonstrated that ICT can support health services in various settings through the use of digital tools for supporting health workers during diagnosis, clinical decision-making, and supervision [20]. The challenges in providing health care are varied in low- and middle-income countries. ICT can address issues such as distance, poor transport infrastructure, and medical provision and provide specialist support in rural areas [18]. Recently, the National Institute for Health and Care Research (NIHR) Global Health Research Unit on Respiratory Health (RESPIRE) included digital technology to improve the diagnosis of pediatric pneumonia [21,22], interpret chest X-rays [23], and to allow home-based pulmonary rehabilitation [24], mHealth support for community health workers [25], remote teleconsultation (micro-health centers) [26] interventions for patients with asthma [27], and blended learning for professionals [28]. To make these interventions efficient and effective, it is necessary to understand ICT infrastructure so that they are accessible to the majority of the population. In Italy, the growing elderly population and chronicity and disability conditions have made necessary the introduction of telecare, particularly in rural areas [18]. Some factors (education, geographical location) can be an obstacle to a successful telemedicine program. A low level of education represents the main difficulty of older people in accessing the internet. In old age, educational qualification represents one of the most decisive factors in the use of new technologies. Statista Research Department in Japan conducted an epidemiological research on internet use in the elderly population, which showed that 57.1% have easy access to internet compared to 75% in USA, and 40.1% in Italy [17,29,30]. Today, in world of advancing technology, telehealth has quickly expanded its goals and abilities. Prior to the COVID-19 pandemic, thousands of physicians and health care professionals provided care to patients in some way with telemedicine [31]. Use of telemedicine has grown due to the COVID-19 pandemic.

Telehealth has modified assistance modalities, reducing hospital patient admissions and reconverting hospital units to accommodate patients affected by COVID-19 in severe clinical conditions.

Remote care services have become essential for patients with disabilities, chronic conditions, and neurological, musculoskeletal, and pain disorders. A majority of participants who use the telemedicine system find these services easy to use, effective, and safe, and are overall satisfied with the new modality of care that they receive. A telephone survey, of patients revealed a variety of benefits, such as alleviating cost- and travel-related burden, the importance of an integrate multidisciplinary model, and effective communication between patients assisted and clinical teams. In addition, they highlighted no difference between telehealth care and traditional face-to-face health care. In particular, teleneurorehabilitation provides a wide range of interventions, such as physiotherapy, speech, cognitive and behavioral therapy, and occupational therapy, as well as telemonitoring and teleconsultation [32]. With the development and evolution of telehealth technologies, the opportunities for patients to access care increase. Telehealth is a highly effective and accessible intervention that can guarantee continuous medical care for patients [18]. In comparison with conventional in-person care, telehealth can eliminate the need to travel distances and reduce crowding in parking areas waiting for lounges [19]. In addition, telehealth helps to reduce travel time and subsequently lower health costs [18]. Moreover, the care provided by telehealth is always performed in home environments. Overall, patients are more satisfied with telemedicine intervention and prefer it to in-person care [18]. It is reported that self-management support and shared decision-making are two effective ways to support PD patients [33,34,35] and for better management of long-term disorders [36,37]. Among the benefits, there is the continuity of care, even after discharge, to the patient’s home, cost savings, improved patient lifestyle, and increased adherence to therapy [37]. Recent studies have shown that the effects of telerehabilitation are comparable to standard care [38,39,40]. In a Carey survey, it emerged that telerehabilitation treatments activate the same cortical regions as conventional treatments [41,42]. On the other hand, many studies have highlighted the need to standardize the procedures, aims, and objectives that characterize this therapeutic modality [18]. Recent literature [43,44,45,46,47] showed that telemedicine in the treatment of PD is valid, even in the most serious pathological states. Most individuals with PD have very limited access to care [31]. Even if initially receiving dedicated and specialist care, the disease’s progression may lead to a homebound status that compromises access to specialized care. For this reason, the use of telemedicine for PD is recommended because it is comparable to in-office visits and feasible, cost-effective, and acceptable to patients [42,43,44,48,49,50]. Considering the efficacy of telerehabilitation, this review aimed to investigate the current use of this innovative tool with PD patients and the efficacy of telerehabilitation on cognitive and psychological outcomes during the COVID era.

## 2. Materials and Methods

### 2.1. Search Strategy

We investigated the impact of telemedicine on cognitive and psychological outcomes in PD patients. The electronic databases PubMed, Web of Science, Scopus, and Cochrane were searched for the period January 2017 to December 2021 using the following terms: “tele[all fields] AND ((“neuro”[all fields] AND “rehabilitation”[all fields]) OR “neuro rehabilitation”[all fields]) AND (“telemedicine”[MeSH Terms] OR “telemedicine”[all fields] OR “telehealth”[all fields]) AND (“parkinson disease”[MeSH Terms] OR (“parkinson”[all fields] AND “disease”[all fields]) OR “parkinson disease”[all fields] OR “parkinson s disease”[all fields]). There was a total of 230 articles identified. Articles were selected by screening the title, abstract, and full text using the following inclusion criteria: (i) original articles; (ii) the study investigated cognitive and psychological symptoms of PD; and (iii) the study applied telerehabilitation in PD patients. We selected only papers in English and removed reviews, case reports, and those on motor outcomes. In total, 13 articles were selected. All studies that examined the cognitive and psychological/psychotherapy treatment of patients with PD by telemedicine were included.

### 2.2. Parkinson’s Disease and Cognitive Telerehabilitation

Cognitive decline is the most common and important nonmotor symptom in PD. In particular, people with PD exhibit more rapid decline in various cognitive domains—executive, attentional, visuospatial, and memory abilities [51]—with prevalence of 25–30%. Given the limited effectiveness of pharmacological treatments on cognitive disorders, nonpharmacological interventions are a valid alternative. Cognitive rehabilitation administered with the support of telemedicine consists of a series of specific exercises aimed at improving cognitive domains, such as attentive processes, memory, spatial cognition, and verbal and nonverbal executive functions. The tool is flexible and adapted to the patient’s abilities. It promotes patient motivation through the playful aspects of the tasks to increase awareness of performance through audio–video feedback [19,52]. From our literary review, it emerged that telerehabilitation, supported by rehabilitative cognitive software, promotes improvement in cognitive deficits, even if the program was managed by remote systems [53]. Indeed, van der Kolk et al. evidenced that home-based aerobic exercise attenuates motor symptoms in Parkinson’s disease and improves nonmotor symptoms. In particular, the authors highlighted the need for larger samples and longer intervention periods to verify long-term effects [53]. Other studies demonstrated that distance is not a barrier to applying neurophysiological techniques combined with cognitive and motor stimulation [47,54]. In a randomized controlled trial, Stillerova et al. reported that monitoring PD symptoms using videoconferencing may be useful to give an overall idea of symptom severity over time [53]. In a randomized crossover pilot study of telemedicine, Sekimoto et al. demonstrated that the outcomes of telemedicine were comparable to those of face-to-face visits, though the authors revealed some limits, such as evaluations that require in-person contact (such as rigidity and postural stability measures) [49,55]. Motor rehabilitation via telemedicine has also demonstrated its efficacy if conducted at home [56]. One study was conducted on a group of idiopathic PD patients. They were supervised online by a therapist, physician, and a designated caregiver to assist the patient. By the end of the telerehabilitation program, the patients had improved motor performance in terms of reduction in risk of falling and improvement in walking speed [57].

Based on these previous findings, Dobbs et al. showed that the use of transcranial direct current stimulation (tDCS) combined with cognitive rehabilitation represented a good method to improve several symptoms, such as cognitive deficits, but also mood disorders, fatigue, and motor deficits. tDCS’s portability, high acceptability, and user-friendly interface make it an easy application for telemedicine practices. Dobbs et al. created a tDCS protocol combined with computerized cognitive rehabilitation for PD patients with improvement in cognitive functions (language, attention, and executive) [55].

### 2.3. Parkinson’s Disease and Telesupport

Mood disorders as well as cognitive disorders are among the nonmotor symptoms of PD [58,59,60,61,62]. In particular, anxiety and depression occur long before diagnosis, as premorbid symptoms are more common in patients with PD than in the general population or those with other chronic conditions [62]. According to Yamanishi et al. [63] and Quelhas and Costa [59], anxiety disorders, more so than depression, have a greater impact on quality of life in PD [64]. Although prevalence rates vary according to sample characteristics and assessment criteria, about one-third of PD patients [65,66] are considered to suffer from clinically relevant depressive and anxiety symptoms. To date, there has been little research into the effectiveness of psychological interventions in PD. Pharmacological interventions for anxiety and depression lead to a high risk of negative side effects and adverse interactions between antidepressant/anxiolytic and antiparkinsonian medications [67,68,69]. Therefore, nonpharmacological interventions can provide a valid alternative for mood problems in PD. In the last 10 years, the development of telemedicine and the necessity of the use of these technologies during the SARS-CoV-2 pandemic have provided evidence of the effectiveness of using telemedicine in the treatment of mood disorders [17,31]. Some studies had already demonstrated the effectiveness of telesupport systems [70,71]. Previous studies [72] showed that telephone support systems were valid tools for anxiety in PD. Ohta et al. created a telephone call service (Andes phone^®^) that was a unique automated support system that called participants who could answer simply by pressing the telephone numbers based on the perceived symptoms. The results suggested that the anxiety levels of the participants and caregivers improved simply by receiving the automated call service every week at home. An automated telephone call service can be a useful care tool for neurological diseases for PD patients living alone or with a caregiver [70,73]. A study conducted on PD and caregivers demonstrated improvements in depression, anxiety, quality of life, sleep, negative thoughts, and caregiver burden independently of treatment modality with guided self-help or formal telephone-based psychotherapy [72]. Dobkin et al. [71] showed that telephone cognitive–behavioral therapy improved depression and anxiety symptoms and quality of life compared to psychological therapy conducted face to face. In particular, cognitive–behavioral therapy combined with usual therapy showed better efficacy than traditional treatment alone. In accordance with Dobkin et al. [72,73,74,75], Wuthrich underlined telephone CBT as a valid tool in the treatment of mood disorders, but it did not significantly reduce anxiety [74,75,76].

The studies analyzed were organized according to the methodology used (Videoconferencing group, Cognitive telerehabilitation- Telephone group, Telesupport), see Table 1.

## 3. Conclusions

Telemedicine is a health-care modality widely tested in clinical practice. With anti-COVID security measures, telemedicine services are essential in reaching patients and their caregivers directly at home. This review aimed to verify the efficacy of telemedicine in rehabilitative treatments of cognitive and psychological deficits typical of PD patients during the COVID era. From the literature, it emerged that telecare represented a useful tool in PD patients. Unfortunately, the samples in these studies were small, making the results not generalizable to the entire population. Most studies focused on rehabilitation of motor symptoms, with few focused on nonmotor symptoms. Among the reasons was the limited availability of PC-based software adaptable to a remote therapy setting to better reach patients at home. Telerehabilitation and telesupport could be good examples of health care. The major techniques of cognitive and psychological telerehabilitation most discussed in the literature were cognitive–behavioral therapy and telecounseling. Future advances could improve and make telehealth-care systems available to weak, older, and low-tech patients, especially PD patients. However, it would be necessary to standardize telemedicine systems for the administration of psychological tests remotely. Guidelines could be created to generalize the results obtained from the application of standard procedures, exploiting the new frontier of artificial intelligence. These literature findings support telemedicine being able to provide an effective care model in the rehabilitation field in patients with neurological disease in terms of monitoring and treating motor and nonmotor symptoms.

In conclusion, our review of the literature supported the development and the efficacy of telerehabilitation, but the quality of the research could be significantly improved to clarify the benefits and risks of telecare.

### Limitations

Telecare systems have different limitations. In particular, the usability of the systems can depend on the type of population. The aging population has more need of assistance in using telemedicine tools and needs support from caregivers. Poor mobile signal coverage limits the availability of mobile/smart devices.

Our review highlighted the lack of standardized procedures for the adaptation of neuropsychological/psychological tests to be administered remotely. Another limitation is that the role of telemedicine in other Parkinson’s-related disorders, such as parkinsonism, has not been addressed in the literature. Future studies may consider related disorders in terms of using telecare systems.

## Figures and Tables

**Table 1 brainsci-13-00499-t001:** Role of telemedicine in the treatment of nonmotor aspects of Parkinson’s disease.

Telemedicine Treatment for Parkinson’s Disease	
Videoconferencing Group Cognitive Telerehabilition	Telephone Group Telesupport
[51] Interactive video conferencing: a means of providing interim care to Parkinson’s disease patients.	[69] Effects of an automated telephone support system on caregiver burden and anxiety: findings from the REACH for TLC intervention study.
[52] Remotely assessing symptoms of Parkinson’s disease using videoconferencing: a feasibility study.	[68] Treatment of behavioural symptoms and dementia in Parkinson’s disease.
[54] Generalizing remotely supervised transcranial direct current stimulation (tDCS): feasibility and benefit in Parkinson’s disease.	[65] Depression in Parkinson’s disease: health risks, etiology, and treatment options.
[53] A randomized crossover pilot study of telemedicine delivered via iPads in Parkinson’s disease.	[70] Telephone-administered cognitive behavioral therapy: a case study of anxiety and depression in Parkinson’s disease.
[49] Effectiveness of home-based and remotely supervised aerobic exercise in Parkinson’s disease: a double-blind, randomized controlled trial.	[76] The relationship between telephone-administered cognitive-behavioral therapy for depression and neuropsychological functioning in Parkinson’s disease.
	[71] Affective improvement of neurological disease patients and caregivers using an automated telephone call service.
	[73] Personalized telemedicine for depression in Parkinson’s disease: a pilot trial.
	[72] Telephone-delivered cognitive behavioural therapy for treating symptoms of anxiety and depression in Parkinson’s disease: a pilot trial.
	[74] Telephone-based cognitive behavioral therapy for depression in Parkinson disease: a randomized controlled trial.
	[75] Innovating Parkinson’s care: a randomized controlled trial of telemedicine depression treatment.

## Data Availability

The data collected was provided by leading databases such as Pubmed, Scopus and Cochrane.

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
