# Peer review of "The Role of Telemedicine in the Treatment of Cognitive and Psychological Disorders in Parkinson’s Disease: An Overview"

_brainsci, 2023, doi:10.3390/brainsci13030499_

Round 1

Reviewer 1 Report

Comments and Suggestions for Authors

This is a timely and relevant review as the literature of current use of telemedicine with Parkinson's disease patients and the efficacy of telerehabilitation on cognitive and psychological outcomes during the Covid era is scarce.

The review needs a major revision as pointed out below:

Line 50-51, 55: Telemedicine became an integral part of non-emergency healthcare [11,12]. Telemedicine is a method of health care, through the use of information and communication technology (ICT).

-Authors should clearly define ‘telemedicine’ as well as ‘telehealth’. Authors are recommended to include the examples of telemedicine/ telehealth. 

Line 67-68: Overall, patients are more satisfied with telehealth intervention and prefer it over in-person care [17].

- This statement is rather broad. Percentage? In which countries/ region? Authors are suggested to provide more precise explanation.

Line 124-125: Already Hubble et al., reported that 124 the motor outcomes could be assessed remotely with a video conference system [41].

-How about the non- motor symptoms including cognition and psychological domain?

Line 141: Parkinson’s disease and tele-support

Under the heading as above; besides telephone consultation, authors should also mention other  aspects of telemedicine being used in the care for people with Parkinson's disease, which includes for example video consultation, virtual multidisciplinary team assessement?, home videos?, werable technology? Remote delivery of treatment?

Line 168- 169: Dobkin et al. [63] showed that telephone cognitive-behavioral therapy enhanced depressive, anxiety symptoms, and quality of life compared to traditional treatment.

- ‘Traditional treatment’ indicates? ‘Pharmocological treatment’ is recommended to replace traditional treatment.

In addition, it would be useful and easier to interpret the data if they are being listed in the table format/ figure.

Thank you

Author Response

Reviewer comment:

This is a timely and relevant review as the literature of current use of telemedicine with Parkinson's disease patients and the efficacy of telerehabilitation on cognitive and psychological outcomes during the Covid era is scarce.

Respose Author: Thank you for the suggestion. Response: The manuscript entitled 'The role of telemedicine in the treatment of cognitive and psychological disorders in Parkinson's disease during the Covid era. An overview' does not focus specifically on Covid although it is present in the title. The literature search also includes many studies carried out during the covid era that were included in the review to substantiate the effectiveness of telemedicine even in a historical period such as the covid era, when therapeutic continuity in patients with Parkinson's disease became necessary. Therefore, we accept the suggestion and remove the term covid from the title to make the work more consistent with the main of the text.

The review needs a major revision as pointed out below:

Line 50-51, 55: Telemedicine became an integral part of non-emergency healthcare [11,12]. Telemedicine is a method of health care, through the use of information and communication technology (ICT).

-Authors should clearly define ‘telemedicine’ as well as ‘telehealth’. Authors are recommended to include the examples of telemedicine/ telehealth. 

Respose Author: I corrected and accepted the suggestion.

Line 67-68: Overall, patients are more satisfied with telehealth intervention and prefer it over in-person care [17].

- This statement is rather broad. Percentage? In which countries/ region? Authors are suggested to provide more precise explanation.

Respose Author: We added information about use of telemedicine in the main countries.

Line 124-125: Already Hubble et al., reported that 124 the motor outcomes could be assessed remotely with a video conference system [41].

-How about the non- motor symptoms including cognition and psychological domain?

Respose Author: I deleted this sentence and integrate this information in the previous sentence.

Line 141: Parkinson’s disease and tele-support

Under the heading as above; besides telephone consultation, authors should also mention other  aspects of telemedicine being used in the care for people with Parkinson's disease, which includes for example video consultation, virtual multidisciplinary team assessement?, home videos?, werable technology? Remote delivery of treatment?

Respose Author: In the previous paragraph we discussed about other use of telemedicine. We added a table with the intervention with telemedicine system in PD patients.

Line 168- 169: Dobkin et al. [63] showed that telephone cognitive-behavioral therapy enhanced depressive, anxiety symptoms, and quality of life compared to traditional treatment.

- ‘Traditional treatment’ indicates? ‘Pharmocological treatment’ is recommended to replace traditional treatment.

Respose Author: I specified the traditional treatment in line 141

In addition, it would be useful and easier to interpret the data if they are being listed in the table format/ figure.

Respose Author: We added a table with telemedicine treatment in Parkinson disease.

Thank you

Reviewer 2 Report

Comments and Suggestions for Authors

This is a review on the possible feasibility of telemedicine in the treatment of cognitive and psychological disorders in Parkinson's Disease (PD). I have the following comments:

1. In the introduction authors should shortly elaborate on the pathogenesis in PD - e.g. inflammatory, genetic etc.

Ref.

Platelet-to-lymphocyte ratio and neutrophil-tolymphocyte ratio may reflect differences in PD and MSA-P neuroinflammation patterns. Neurol Neurochir Pol. 2022;56(2):148-155. doi: 10.5603/PJNNS.a2022.0014. Epub 2022 Feb 4. PMID: 35118638.

 The Genetic Testing Experience of Individuals with Parkinson's Disease. Mov Disord Clin Pract. 2023 Jan 2;10(2):248-257. doi: 10.1002/mdc3.13641. PMID: 36825058; PMCID: PMC9941910.

2. In the material section authors mentioned analyzing text before COVID era - from 2017. In this context I believe that the title should indicate the role of telemedicine not only in COVID era.

3. The role of telemedicine seems more interesting in the context of parkinsonisms with limited mobility. It would be valuable to mention the role of telemedicine in related diseases - DLB/PSP etc.

4. The limitations of telemedicine should be indicated in. a separate paragraph

5. Authors could add a table summarizing the added value of telemedicine and weaknesses. This should be accompanied by a critical point of view based on the authors' opinions.

Author Response

Reviewer comment:

This is a review on the possible feasibility of telemedicine in the treatment of cognitive and psychological disorders in Parkinson's Disease (PD). I have the following comments:

  1. In the introduction authors should shortly elaborate on the pathogenesis in PD - e.g. inflammatory, genetic etc.

Ref.

Platelet-to-lymphocyte ratio and neutrophil-tolymphocyte ratio may reflect differences in PD and MSA-P neuroinflammation patterns. Neurol Neurochir Pol. 2022;56(2):148-155. doi: 10.5603/PJNNS.a2022.0014. Epub 2022 Feb 4. PMID: 35118638.

 The Genetic Testing Experience of Individuals with Parkinson's Disease. Mov Disord Clin Pract. 2023 Jan 2;10(2):248-257. doi: 10.1002/mdc3.13641. PMID: 36825058; PMCID: PMC9941910.

Response Author: I added references that you suggested and specified pathogenesis in the introduction section

  1. In the material section authors mentioned analyzing text before COVID era - from 2017. In this context I believe that the title should indicate the role of telemedicine not only in COVID era.

Response Author: in the material section we described that our research is from January 2017 to December 2021. This review not focus specifically on Covid although it is present in the title. The literature search also includes many studies carried out during the covid era that were included in the review to substantiate the effectiveness of telemedicine even in a historical period such as the covid era, when therapeutic continuity in patients with Parkinson's disease became necessary. We removed the term covid from the title to make the work more consistent with the main of the text.

  1. The role of telemedicine seems more interesting in the context of parkinsonisms with limited mobility. It would be valuable to mention the role of telemedicine in related diseases - DLB/PSP etc.

Response Author:  Thank you for this suggestion. We conducted a research on pubmed with the keyword “telemedicine in parkinsonisms” and “role of telemedicine in Dementia with Lewy Bodies” and “role of telemedicine in progressive supranuclear palsy” there are no studies in this field. We added the lack of studies in these related diseases in limitation section.

  1. The limitations of telemedicine should be indicated in. a separate paragraph

Response Author: I added limitations paragraph

  1. Authors could add a table summarizing the added value of telemedicine and weaknesses. This should be accompanied by a critical point of view based on the authors' opinions.

Response Author: We added a table summerizing telemedicine treatment in Parkinson Disease.

Round 2

Reviewer 1 Report

Comments and Suggestions for Authors

The authors had revised the manuscript accordingly.

Just a small correction for the legend of the table:

''Table 1. Telemedicine tretments in Parkinson Disease''

Treatment is spelled wrongly.

It should be written as Role of telemedicine in the treatment of non-motor aspect of Parkinson's  disease., as this overview mainly highlighted the use of telemedicine in cognitive and psychological aspect of PD.

Author Response

thank you, for your observations. I corrected the legend of the table as you suggested. 

Reviewer 2 Report

Comments and Suggestions for Authors

Authors implemented my comments.

Minor comment:

"But, it remains un- 31 clear, if the prolonged inflammation in the cause or an effect of neurodegeneration [2]" - I believe there is a spelling mistake here

Author Response

Thank you,

I corrected the spelling mistake as you suggested.